# Learning to Refer to 3D objects with Natural Language

## Abstract

Human world knowledge is both structured and flexible. When people see an object, they represent it not as a pixel array but as a meaningful arrangement of semantic parts. Moreover, when people refer to an object, they provide descriptions that are not merely true but also relevant in the current context. Here, we combine these two observations in order to learn fine-grained correspondences between language and contextually relevant geometric properties of 3D objects. To do this, we employed an interactive communication task with human participants to construct a large dataset containing natural utterances referring to 3D objects from ShapeNet in a wide variety of contexts. Using this dataset, we developed neural listener and speaker models with strong capacity for generalization. By performing targeted lesions of visual and linguistic input, we discovered that the neural listener depends heavily on part-related words and associates these words correctly with the corresponding geometric properties of objects, suggesting that it has learned task-relevant structure linking the two input modalities. We further show that a neural speaker that is 'listener-aware' — that plans its utterances according to how an imagined listener would interpret its words in context — produces more discriminative referring expressions than an 'listener-unaware' speaker, as measured by human performance in identifying the correct object.

## 1 Introduction

Human world knowledge is both structured and flexible. For example, when people see a chair, they represent it not as a pixel array but as a semantically meaningful combination of parts, such as *arms, legs, seat,* and *back*. How to obtain and flexibly deploy such structured knowledge remains an outstanding problem in machine learning (Lake et al., 2017). One promising approach is to harness the rich conceptual and relational structure latent in language (Andreas et al., 2017). Natural languages have been optimized across human history to solve the problem of efficiently communicating those aspects of the world most relevant to current goals (Kirby et al., 2015; Gibson et al., 2017). Consequently, language reflects the structured nature of our world knowledge: we not only conceive of a chair in terms of its semantic parts, but can combine multiple words to refer to its 'curved back' or 'cushioned seat', and provide more informative descriptions if the context requires it, e.g., refer to a different distinguishing part if all the chairs have a cushioned seat.

Our goal is to leverage these insights to develop systems that can make fine-grained distinctions between complex object geometries across a wide variety of contexts. Our approach is to leverage natural language produced by people in an interactive communication task to develop neural network models of the speaker and listener roles in this task. We find that the resulting representations learned by these models exhibit structure that is crucial for robust communication: first, they capture task-relevant correspondences between individual parts of objects and individual tokens of language, and second, they have strong capacity to generalize to novel contexts, objects, utterances, and other related object classes.

We make the following contributions:

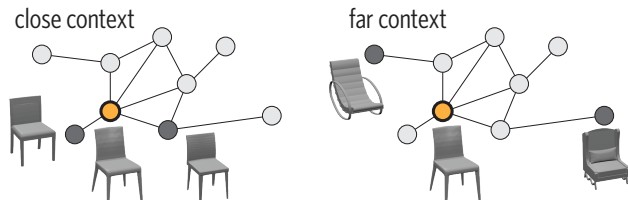

Figure 1: Constructing "close" and "far" contexts by exploiting the latent neighborhood structure of 3D chairs. Orange is a high in-degree seed chair, dark gray its selected distractors in each context.

- We introduce a new multimodal dataset (*Chairs In Context*) comprised of 4,511 chairs from ShapeNet, organized into 4,054 sets of size 3 (called communication contexts), with 78,789 natural utterances, each utterance intended to distinguish a chair in context.[1]

- By training on this dataset we develop neural listeners and speakers with strong generalization capacity even in *out-of-training* classes, such as tables.

- We demonstrate that the neural listener learns to prioritize the same geometric information in objects (i.e., properties of individual chair parts) that humans do in solving the communication task, despite never being provided with an explicit decomposition of these objects into parts.

- We show how our listeners can be used to search large collections of *unseen* objects to retrieve models based on natural language queries, e.g., *curved back and fat legs*.

- Lastly, we find that a neural speaker that is 'listener-aware' — that plans its utterances according to how an imagined listener would interpret its words in context — produces more discriminative utterances than an 'listener-unaware' speaker, as measured by human performance in identifying the correct object.

## 2   DATASET AND TASK

Our dataset consists of triplets of 3D objects coupled with referential utterances that aim to distinguish one object (the "target") from the remaining two (the "distractors"). To obtain such utterances, we paired participants from Amazon Mechanical Turk to play an online, *reference game* (Hawkins, 2015). On each round of the game, the two players were shown a triplet of objects. The designated target object was privately highlighted for one player (the "speaker") who was asked to send a message through a chat box such that their partner (the "listener") could successfully select it from the context (see Appendix Fig. 16). To ensure speakers used *geometric* information rather than color, texture, orientation, or position on the screen, we scrambled the positions of the objects for each participant and used textureless, colorless renders of 3D objects taken from the same viewpoint. Additionally, to ensure communicative interaction was natural, no constraints were placed on the chat box: referring expressions from the speaker were occasionally followed by clarification questions from the listener or other discourse.

A crucial decision in building our dataset concerned the construction of useful contexts that would reliably elicit *fine-grained* contrastive language. Perceptually identical objects cannot be distinguished with language at all, while wildly different objects (a chair and a car) can be easily distinguished with a single word ("it's a chair"). To solve this problem, we considered three objectives. First, the set of objects must be familiar so we can tap existing visual and linguistic representations. Second, the objects should be complex and variable to provide wide coverage of interesting geometries. Third, different contexts must contain diverse *combinations* of objects to ensure variation in the relevant distinctions required.

To satisfy the first two objectives, we utilize the collection of about 7,000 chairs from ShapeNet (Chang et al., 2015). This class is geometrically complex, densely sampled, highly diverse, and abundant in the real world. To satisfy the third objective in a scalable and unsupervised manner, we estimated object similarity between different chairs using the Point Cloud-AutoEncoder (PC-AE) from Achlioptas et al. (2018). This representation allowed us to leverage the fact that point-clouds extracted from a 3D surface provide an intrinsic 3D representation of an object, oblique to color or texture. To deal with the inhomogeneity of data in repositories like ShapeNet we used a sampling

---

[1]The dataset and our code will be publicly available upon acceptance.

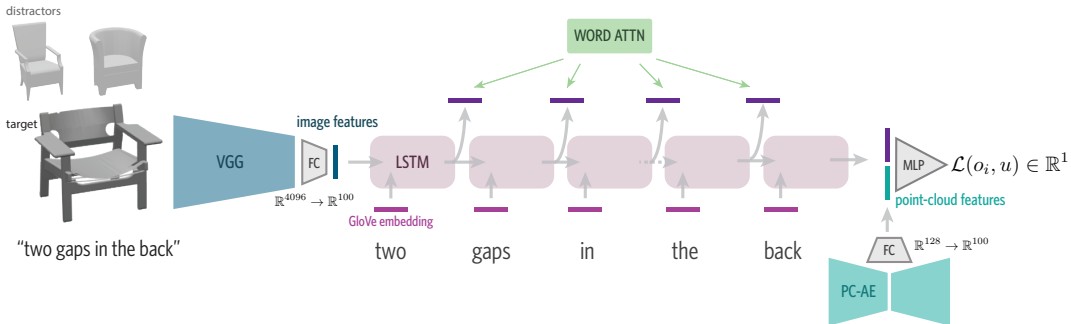

Figure 2: Proposed listener architecture.

strategy to construct our triplets. First, we computed the 2-nearest-neighbor graph of all ShapeNet chairs based on their PC-AE embedding distances. On this graph, we selected chairs of highest in-degree as seeds and for each seed chair we generated two kinds of triplets. *Close* contexts sampled nearby chairs while *Far* contexts sampled highly dissimilar chairs (see Fig. 1). Additional details of triplet construction are provided in the Appendix (Section 9.1).

In total, we collected a corpus containing 78,789 referring expressions for 4,054 triplets, containing 4,511 unique chairs. In doing so we recruited 2,124 unique participants. Human performance on the reference game was high in general, but listeners made significantly more errors in the close triplets (94.2% vs. 97.2%, $z = 13.54, p < 0.001$). Also, significantly longer utterances were used on average to describe targets in close triplets (approximately 8.4 words vs. 6.1, $t = -35, p < 0.001$). A wide spectrum of descriptions was elicited, ranging from the more holistic/categorical common for far triplets (e.g., "the rocking chair") to more complex, geometric language common for close triplets (e.g., "thinner legs but without armrests"). 78% of utterances used at least one part word: "back", "legs", "seat," "arms", or closely related synonyms (e.g., "armrests").

## 3 NEURAL LISTENERS

Constructing neural listeners that reason about geometric relationships is a key contribution of our work. It lays the foundation for creating speakers that utter discriminative utterances and enables the creation of an object retrieval system that operates with linguistic queries. Given its importance, below we conduct a detailed comparison between three distinct architectures, highlight the effect of different regularization techniques, and investigate the merits of two different representations of 3D objects for the listening task, namely, images and point clouds. In what follows, we denote the three objects of a communication context as $O = \{o_1, o_2, o_3\}$, the corresponding word-tokenized utterance, which has at most $K$ tokens , as $U = u_1, u_2, \ldots$ and as $t \in O$ the referential target.

Our proposed listener is inspired by Monroe et al. (2017). It takes as input a (latent code) vector for each of the three objects in $O$ and a (latent code) vector for each token of $U$, and outputs an object–utterance compatibility score $\mathcal{L}(o_i, U) \in [0, 1]$ for each of the three objects. At its core lies a multi-modal LSTM (Hochreiter & Schmidhuber, 1997) that takes as input ("is grounded" with) the vector of a single object, processes the word-sequence $U$, and is read out by a final MLP to yield a single number (the compatibility score). This is repeated for each $o_i$, sharing all network parameters across the objects. We then apply a soft-max to the three compatibility scores to yield a distribution over the three objects, and compute a cross-entropy loss between this distribution and the ground-truth indicator vector of the target.

**Object encoders** We experimented with three object representations to capture the underlying geometries: (a) the bottleneck representation of a pretrained Point Cloud-AutoEncoder (PC-AE), (b) the embedding provided by a convolutional network operating on single-view images of non-textured 3D objects, or (c) a combination of (a) and (b). Specifically, for (a) we use the PC-AE architecture of Achlioptas et al. (2018) trained with single-class point clouds extracted from the surfaces of 3D CAD models, while for (b) we use the activations of the penultimate layer of a VGG-16 (Simonyan & Zisserman, 2014) neural network, pre-trained on ImageNet (Deng et al., 2009), and fine-tuned on an 8-way classification task with images of objects from ShapeNet. For each

representation we project the corresponding code vector to the input space of the LSTM using a fully connected (FC) layer with $L_2$-norm weight regularization. The addition of these projection-like layers improves the training and convergence of our system.

While there are many ways to simultaneously incorporate the two modalities in the LSTM, we found that the best performance resulted when we ground the LSTM with the image code, concatenate the LSTM's final output (after processing $U$) with the point cloud code, and finally feed this result into a shallow MLP to produce the final compatibility (see Figure 2 for an overview of this architecture). We note that grounding the LSTM with point clouds and using images towards the end of the pipeline, resulted in a significant performance drop ($\sim 4.8\%$ on average). Also, adding dropout at the input layer of the LSTM and $L_2$ weight regularization and dropout at and before the FC projecting layers was crucial (giving improvements of more that $10\%$). The token codes of each sentence where initialized with the GloVe embedding (Pennington et al., 2014) and fine-tuned for the listening task.

**Incorporating context information**   Critically, our proposed listener architecture first scores each object *separately* then applies softmax normalization to yield a score distribution over the three objects. In order to evaluate the importance of this design choice, we consider two alternative architectures that incorporate context earlier, at encoding. The first alternative (*Separate-Augment*), is identical to the proposed architecture, except for it uses a convolutional layer to augment each object's grounding vector with information about the other two objects in context before yielding its score. Specifically, if $v_i$ is the image code vector of the i-th object ($o_i \in O$), to produce the grounding vector for $o_i$, the convolutional layer receives $f(v_j, v_k)||g(v_j, v_k)||v_i$, where $f, g$ are order invariant functions, such as the average or max-pooling and $||$ denotes feature-wise concatenation. The second alternative architecture (*At-Once*) first feeds the image vectors for all three objects sequentially to the LSTM and then proceeds to process the tokens of $U$ *once*, to yield the entire score distribution. Similarly to the proposed architecture, point clouds are incorporated in both alternatives via a separate MLP after the LSTM.

**Attention mechanism over words**   We hypothesize that a listener forced to prioritize a few words in each utterance would learn to prioritize words that express properties that distinguish the target from the distractors (and, thus, perform better). To test this hypothesis, we augment the listener models with a *bilinear attention mechanism* over words. Specifically, to estimate the "importance" of each text-token $u_i$ we compare the output of the LSTM for $u_i$ (denoted as $r_i$) with the hidden state after the entire utterance has been processed (denoted as $h$). The ideas is that the hidden state acts as a summary of the grounded sentence (Shen & Lee, 2016), that can be used to assess the relative importance of each word as $a_i \triangleq r_i^T \times W_{att} \times h$, where $W_{att}$ is a trainable diagonal matrix. With the attention mechanism in place, the final output of the LSTM is defined as $\sum_{i=1}^{|U|} r_i \odot \hat{a}_i$, where $\hat{a}_i = \frac{\exp(a_i)}{\sum_j^{|U|} \exp(a_j)}$ and $\odot$ is the point-wise product.

The optimal parameters of each listener (and speaker), the hyper-parameter search strategy, and the exact details of training are provided in the Appendix 9.2 and 9.3.

## 4   NEURAL SPEAKERS

**Architecture**   Our speaker models are inspired by the show-and-tell model (Vinyals et al., 2015) developed for image captioning. Specifically, a speaker is a neural network that receives an image-based code vector per object in $O$ and learns to generate an utterance $U$ that refers to the target and which distinguishes it from the distractors. Similarly to the listener model, the main components of the speaker's architecture are an LSTM and a convolutional image network (we do not include point clouds when speaking to allow for a more easily deployable model). During the first three time steps, the speaker receives sequentially the three image code vectors of a context (projected via an $L_2$-norm weight regularized FC) and outputs a vector which is transformed into a logit prediction over our vocabulary via an FC.

The soft-normalized version of the output is compared against the first ground-truth token ($u_1$) under the cross-entropy loss. For each remaining token $u_i \in u_2, \ldots$, the LSTM is conditioned on the previous ($u_{i-1}$) ground-truth token and the cross-entropy comparison is repeated (i.e., we do teacher-forcing (Williams & Zipser, 1989)). In all speakers the target vector is fed third,

Table 1: Performance of variants of proposed listener architecture (image-modality, attention, and context-incorporation alternatives) on different generalization tasks and subpopulations of the test set (chance is 33%; mean accuracy $\pm 1$ standard error). Bottom table uses best-performing model from top table. Averages taken over five random seeds that controlled the data splits and neural-net initializations.

| | Input-Modality | Language-Task | Object-Task |
|---|---|---|---|
| **No Attention** | Point Cloud | $67.6 \pm 0.3\%$ | $66.4 \pm 0.7\%$ |
| | Image | $81.2 \pm 0.5\%$ | $77.4 \pm 0.7\%$ |
| | Image & Point Cloud | $83.1 \pm 0.2\%$ | $78.9 \pm 1.0\%$ |
| **With Word-level Attention** | Point Cloud | $67.4 \pm 0.3\%$ | $65.6 \pm 1.4\%$ |
| | Image | $81.7 \pm 0.5\%$ | $77.6 \pm 0.8\%$ |
| | Image & Point Cloud | $\mathbf{83.7} \pm 0.2\%$ | $\mathbf{79.6} \pm 0.8\%$ |

| Architecture | Subpopulations | | | |
|---|---|---|---|---|
| | Overall | Close | Far | Sup-Comp |
| At-Once | $75.9 \pm 0.5\%$ | $67.4 \pm 1.0\%$ | $83.8 \pm 0.6\%$ | $74.4 \pm 1.3\%$ |
| Separate-Augment | $79.4 \pm 0.8\%$ | $\mathbf{70.1} \pm 1.3\%$ | $\mathbf{88.1} \pm 0.6\%$ | $75.2 \pm 2.1\%$ |
| Separate (proposed) | $\mathbf{79.6} \pm 0.8\%$ | $69.9 \pm 1.3\%$ | $\mathbf{88.1} \pm 0.4\%$ | $\mathbf{76.0} \pm 1.6\%$ |

thereby minimizing the length of dependence between the most important input object and the output (Sutskever et al., 2014) and eliminating the need to represent the index of the target separately. To find the best hyper-parameters ($L_2$ weights, dropout-rate and # of LSTM neurons) and the optimal (per validation) epoch, during training we sample synthetic utterances of each model and use a pretrained *listener* to select the combination with the highest listener accuracy. We found this approach to produce model parameters with stronger correlation between the training 'progress' and the quality of produced utterances, than using listening-unaware metrics like BLEU (Papineni et al., 2002).

**Variations** In principle, the above speaker can learn to generate language that follows the discriminative characteristics of the referential ground truth. To test the degree to which the distractors are taken into account for this purpose, we experiment with a speaker that is "context-unaware" by construction. This speaker at both training and test time uses the image encoding of the target object *only*, and is otherwise identical to the above model. Then, motivated by the recursive social reasoning formalized in the Rational Speech Act framework (Goodman & Frank, 2016), we create a *listener-aware* speaker that plans synthetic utterances according to their capacity to be discriminative, as judged by an "internal" listener. In this case, a speaker's *sampled* utterance $U$ is scored as:

$$\text{score(U)} = \beta \log(P_L(t|U)) + (1 - \beta) \sum_{i=1}^{i=|U|} \frac{\log(P_S(u_i|O))}{|U|^\alpha} \ , \tag{1}$$

where $P_L$ is the listener's probability to predict the target ($t$) given $U$, $u_k$ is a token of $U$ and $P_S$ is the likelihood of the speaker for generating $U$ given the objects in $O$. The parameter $\alpha$ controls a length-penalty term to discourage short sentences (Wu et al., 2016), while $\beta$ controls the relative importance of the speaker's vs. the listener's opinions.

## 5 LISTENER EXPERIMENTS

We evaluated our listener's generalization performance using two tasks based on different data splits. In the *language generalization* task, we test on target objects that were seen as targets in at least one context during training but ensured that all utterances in the test split are from unseen speakers. In the more challenging *object generalization* task, we restricted the set of objects that appeared as targets in the test set to be *disjoint* from those in training such that all objects *and* utterances in the test split are unseen. For each of these tasks, we evaluated choices of input modality and word attention, using $[80\%, 10\%, 10\%]$ of the data, for training, validation and test for all experiments. Listener accuracies are shown in Table 1 (top).

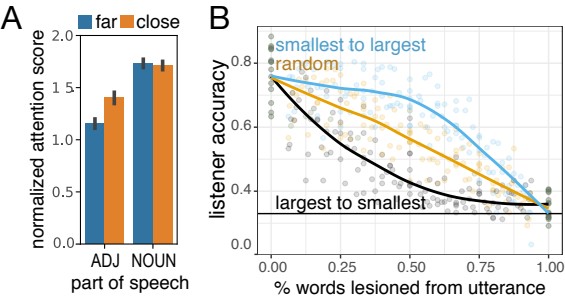

Figure 3: (A) The listener places more attention on adjectives in close (orange) triplets than far (blue) ones. (B) Lesioning highest attention words to lowest worsens performance more than lesioning random words or lesioning lowest attention words.

First, as expected, all architectures have higher accuracy on the language generalization task (3.2% on average). Second, the attention mechanism on words yields a mild performance boost, as long as images are part of the input. Third, images provide a much better input than point-clouds when only one modality is used. In other words, despite being an intrinsic 3D representation, point-clouds alone seem to provide a weaker input signal, perhaps due to their relative lack of high-frequency details. Finally, we find large gains in accuracy (4.1% on average) from exploiting the two modalities simultaneously, potentially implying a complementarity between the two representations that the network can exploit.

Next, we evaluate how the different approaches to incorporating context information described in Section 3 affect listener performance. We focus on the more challenging object generalization task, using models that include attention and both object modalities. See Table 1 (bottom) for results. We find that the At-Once architecture, which consumes the entire context with a single (non-weight shared replica) LSTM, performs significantly worse than both the Separate and Separate-Augment architectures (which use an explicit shared-weighting mechanism) and achieve similar performance to each other. It is plausible that our alternative strategies for incorporating context information however, would yield an advantage in close contexts, where finer distinctions must be made. However, we did not observe differences between the Separate and Separate-Augment variants in either the far *or* the close subpopulations; we do find that far contexts were easier for all models than close contexts. Surprisingly, we found that the Separate architecture remains competitive against the Separate-Augment architecture even in the subpopulation that includes utterances with superlatives and/or comparatives ("skinnier"/"skinniest") which made up $\sim 16\%$ of the test set and make explicit reference to context. Since the Separate architecture is the most flexible (see Section 5.2 for a demonstration of this), and is also simpler than Separate-Augment while performing equally well, we focus on this in the following sections.

## 5.1 EXPLORING LEARNED REPRESENTATIONS

Which aspects of a sentence are more critical for our listener's performance? To inspect the properties of words receiving the most attention, we ran a part-of-speech tagger on our corpus. We found that the highest attention weight is placed on *nouns*, controlling for the length of the utterance. However, adjectives that *modify* nouns received more attention in close contexts (controlling for the average occurrence in each context), where nouns are often not sufficient to disambiguate (see Fig. 3A). To more systematically evaluate the role of higher-attention tokens in listener performance, we conducted an utterance lesioning experiment. For each utterance in our dataset, we successively replaced words with the <UNK> token according to three schemes: (1) from highest attention to lowest, (2) from lowest attention to highest, and (3) in random order. We then fed these through an equivalent listener trained *without* attention. We found large differential performance from random in both directions (see Fig. 3B). This ablation result was found across a wide range of utterance lengths. Our word-attentive listener thus appears to rely on context-appropriate content words to successfully disambiguate the referent. Examples demonstrating where the attention is being placed on utterances produced by *humans* are given in Appendix Fig. 7.

To test the extent to which our listener is relying on the same semantic parts of the object as humans, we conducted a lesion experiment on the visual input rather than the linguistic one. We took the subset of our test set where (1) all chairs had complete part annotations available (Yi et al., 2016) and (2) the corresponding utterance mentioned a single part (17.5% of our test set). We then rendered

Table 2: Testing the part-awareness of neural listener by lesioning different parts of the objects. Reported is the average accuracy of a listener under different lesions.

|  | Intact Object | Single Part Lesioned | Single Part Present |
|---|---|---|---|
| **Mentioned Part** | 78.5% | $41.8\% \pm 0.1$ | $66.6\% \pm 0.1$ |
| **Random Part** | | $67.0\% \pm 0.2$ | $37.4\% \pm 0.1$ |

Figure 4: Top-scoring retrieved results in collections of unseen objects with natural-language queries. Bottom two rows include *out-of-class* examples from collections of lamps, sofas and tables.

lesioned versions of all three objects on each trial by removing pixels corresponding to parts [2] according to two schemes: *removing* a single part or *keeping* a single part. We did this either for the mentioned one, or another part, chosen at random. We report listener accuracies on these lesioned contexts in Table 2. We found that removing random parts hurts the accuracy by 11% on average, but removing the mentioned part dropped accuracy more than three times as much, nearly to chance. Conversely, keeping only the mentioned part while lesioning the rest of the image merely drops accuracy by 11.9% while keeping a non-mentioned (random) part alone brings accuracy down to 37.4% on average. In other words, on trials when participants depended on information about a part to communicate the object to their partner, we found that *localized* information about that part was both necessary and sufficient for the performance of our listener model.

## 5.2 USING LISTENER FOR RETRIEVAL IN NOVEL OBJECT COLLECTIONS

Finally, as a demonstration the broader applicability of our listener, we consider the problem of searching a large database of 3D objects using natural language queries. A key advantage of the proposed listener is its flexibility to be applied on arbitrary sized contexts. We exploit this flexibility by using a pre-trained listener to measure the compatibility $\mathcal{L}(o_i, U)$ between *every* object of a test collection $O = \{o_1, \ldots o_N\}$ and the query $U$. In Figure 4 (top) we show the chairs of the held-out splits (a set of 900 chairs) with the highest compatibility for a range of utterance queries. Additionally, we show the results of applying this model (trained on chairs) in the entire *out-of-training* classes of (ShapeNet) sofas, tables and lamps (object sets of size 3.2K, 8.5K, and 2.3K, respectively). We see surprisingly good results on searching these transfer categories. This further supports the part-awareness of the learned embedding, since the commonality between a chair and a table can be primarily expressed trough their shared parts. In the Appendix we include additional queries for chairs (Fig. 10) and non-chairs (Fig. 11).

## 6 SPEAKER EXPERIMENTS

Having established that our neural listener learns useful representations with surprisingly structured properties, we now proceed to evaluate our neural speakers (see Fig. 5 for examples). [3]

We evaluate them by measuring their success in referential games with two different kinds of partners: with an independently trained listener model and with human listeners on Amazon

---

[2]Due to the added difficulty of annotating and lesioning point-clouds, we conducted this experiment with the image-only variant of our listener

[3]Appendix Fig. 12 illustrates how the speakers refer to the same targets in far vs. close contexts. Appendix Figs 9 and 8 showcase listener's and speaker's failure modes.

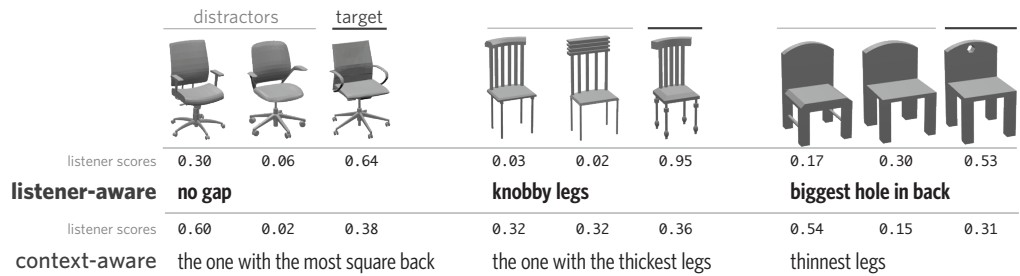

Figure 5: Top-scoring synthetic utterances generated from listener-aware and context-aware speakers for *unseen* targets. Proportions correspond classification scores of our independent evaluating listener.

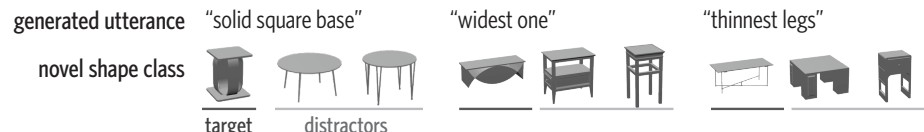

Figure 6: Our listener-aware speaker can produce informative referring expressions for *out-of-class* objects in context. Here, we apply our search technique in the collection of ShapeNet Tables, to produce triplets of well-separated objects. We use the queries: 'no legs' (left), 'modern' (center), and 'x' (right), to construct each triplet. Notice that the target of each triplet (selected from the highest-ranked matches) reflects the semantics of the used query, as opposed to the distractors (selected from the lowest-ranked matches).

Mechanical Turk (see Table 3). Critically, to conduct a fair evaluation using a neural listener, we split our training data in half. The evaluating listener was trained using one half while the scoring (or "internal") listener used by the speaker to choose utterances was trained on the remaining half. For our human evaluation, we used the context- and listener-aware speakers to generate synthetic referring expressions on the test set. To avoid data-spillage we use all training data to train the internal listeners here. We then showed these referring expressions to participants and asked them to select the object from context that the speaker was referring to. We collected approximately 2.58 responses for each triplet. For all speaking experiments, we used the same object-generalization splits used in the listening experiments. The synthetic utterances were the best scoring sentence according to each model with optimal $\alpha$ and a subset of $\beta$ values (See Appendix Section 12 for the effect of these hyper-parameters in a wider range).

We found that our listener-aware speaker, which uses an internal listener model to produce informative utterances, performs significantly better in reference games. While its success with the evaluating listener model may be unsurprising, given the architectural similarity of the internal listener and the evaluating listener, *human* listeners were 10.4 percentage points better at picking out the target on utterances produced by the listener-aware speaker. While for listeners we found it was sufficient to bring context into the model only at the final stage, the soft-max over objects, we found that for speakers it was helpful to bring context into play earlier: the context-unaware speaker does significantly worse than context-aware one (64.0% vs. 76.6%). Qualitatively, we note that both context/listener aware speakers produce *succinct* descriptions (average sentence length

Table 3: Speaker evaluations. For the neural listeners, five random seeds controlling the weight initialization and speaker-listener data splits were used.

| Speaker-Architecture | Listener Model | Human Listeners |
|---|---|---|
| Context-unaware | $64.0 \pm 1.7\%$ | - |
| Context-aware ($\beta = 0.0$) | $76.6 \pm 1.0\%$ | 68.3 |
| Listener-aware ($\beta = 0.5$) | $85.9 \pm 0.4\%$ | - |
| Listener-aware ($\beta = 1.0$) | $92.2 \pm 0.5\%$ | 78.7 |

4.21 vs. 4.97) but the listener-aware speaker uses a much richer vocabulary (14% more unique nouns and 33% more unique adjectives, after controlling for average length discrepancy). As a final qualitative examination of our speakers' generalization ability, we ran a simple *out-of-class* speaking experiment. We constructed well-separated contexts from the search results presented in Section 5.2, taking as the target the highest-ranked exemplar and choosing distractors from among the lowest-ranked. Our best speaker model produced promising results (see Fig. 6).

# 7 RELATED WORK

**Image labeling and captioning**     Our work builds on recent progress in the development of vision models that involve some amount of language data, including object categorization (Simonyan & Zisserman, 2014; Zhang et al., 2014) and image captioning (Karpathy & Fei-Fei, 2015; Vinyals et al., 2015; Xu et al., 2016). Unlike object categorization, which pre-specifies a fixed set of class labels to which all images must project, our system uses open-ended, natural language. Similarly to other recent works in image captioning (Luo & Shakhnarovich, 2017; Monroe et al., 2017; Vedanta et al., 2017) instead of captioning a single image in isolation, our systems learn how to communicate across diverse semantic *contexts*. More importantly, using 'clean' images of separate articulated objects enables the generation of very fine-grained, part-based descriptions.

**Reference games**     In our work we use reference games in order to operationalize the demand to be relevant in context. The basic arrangement of such games can be traced back to the language games explored by Wittgenstein (Wittgenstein, 1953) and Lewis (Lewis, 1969). For decades, such games have been a valuable tool in cognitive science to quantitatively measure inferences about language use and the behavioral consequences of those inferences (Rosenberg & Cohen, 1964; Krauss & Weinheimer, 1964; Clark & Wilkes-Gibbs, 1986; van Deemter, 2016). Recently, these approaches have also been adopted as a benchmark for discriminative or context-aware NLP (Paetzel et al., 2014; Andreas & Klein, 2016; Cohn-Gordon et al., 2018; Vedantam et al., 2017; Su et al., 2017; Lazaridou et al., 2018).

**Rational Speech Acts framework**     Rational Speech Act (RSA) models provide a probabilistic framework for deriving linguistic behavior from general principles of social cognition (Goodman & Frank, 2016). At the core of the RSA framework is the Gricean proposal (Grice, 1975) that speakers are decision-theoretic agents who select utterances $u$ that are parsimonious yet informative about the state of the world $w$. RSA formalizes this notion of informativity as the expected reduction in the uncertainty of an (internally simulated) listener $L$: $S(u|w) \propto \exp\{\log L(w|u)\}$, $L(w|u) \propto \mathcal{L}(u, w)$. This speaker $S$ is *pragmatic* because it considers informativity in terms of a rational listener agent ($L$) who updates their beliefs about the world according to the literal semantics of the language ($\mathcal{L}$). Previous work has shown that RSA models account for context sensitivity in human speakers (Graf et al., 2016; Monroe et al., 2017; Yu et al., 2017; Fried et al., 2017). Our speaking results add evidence in the effectiveness of this approach.

# 8 CONCLUSION AND FUTURE DIRECTIONS

Taken together, our results show that natural language, derived from communication in context, provides a strong objective for learning to make fine-grained distinctions between objects with an emphasis on their shared part-structure. An exciting future application of this work would be to leverage these techniques for improving unsupervised part segmentation and 3D shape retrieval, as well as context-aware shape synthesis, providing an advance over existing context-unaware synthesis techniques (Chen et al., 2018).

ACKNOWLEDGMENTS

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

## 9 APPENDIX

### 9.1 DETAILS ON BUILDING THE CONTEXTS

To build our contrastive triplets, we first compute the 2-nearest-neighbor graph of all ShapeNet chairs based on their Euclidean latent distances and use a subset of 1K chairs, those with the highest in-degree on this graph to "seed" the triplet generation. Given a node of this graph, we select its two nearest neighbors from the *entire* shape collection to form a triplet of highly similar ("*close*") objects and also select the two objects that are closest to it but which also are more distant from it than the median of all pairwise distances, to form a triplet of relatively "*far*" objects. Having two types of contexts (close and far) allowed us to collect contrastive language with various degrees of specificity. To counterbalance the dataset we ensured that each object of a triplet alternated roles (with the remaining two) as a distractor or target, and that each resulting combination was annotated by at least 4 humans. The AE used to make the embedding was of a relative small size (64D) to promote meaningful Euclidean comparisons. Also, for the close triplets we applied a manually tuned threshold, to semi-automatically reject triplets that contained two indistinguishable geometric objects, e.g., two that were listed in ShapeNet but only varied on their texture.

### 9.2 LISTENERS DETAILS

| Param/Architecture | Proposed | At-Once | Separate-Augment |
|---|---|---|---|
| Learning-rate | 0.0005 | 0.001 | 0.001 |
| Label-smoothing | 0.9 | 0.9 | 0.9 |
| $L_2$-reg. | 0.3 | 0.05 | 0.09 |
| RNN-dropout | 0.5 | 0.7 | 0.45 |

Table 4: Optimal hyper parameters for neural listener architectures using both images and point clouds and word attention.

The optimal values for the hyper-parameters used by each listener model (using both point clouds and images and word-attention) are given in Table 4. All listeners use an MLP with [100, 50] hidden neurons (FC-ReLU (Maas et al., 2013)) with batch normalization after each layer and an LSTM with 100 hidden units. The GloVe embedding was also 100-dimensional and it was fine-tuned during training. For the point-cloud latent bottleneck codes (128D) and the VGG image features (4096D) we use a dropout with 0.5 keep probability to zero half their entries *before* using the FC-projecting layers. The same drop-out mask was applied on the codes of a given triplet. The ground-truth indicator vectors were label-smoothed (Szegedy et al., 2015). Assigning a probability of 0.933 to the ground-truth target and 0.0333 to the distractors (smoothing of 0.9, second row of Table 4) yielded a performance boost of $\sim 2\%$. Label-smoothing has been found also in previous work to improve the generalization (Szegedy et al., 2015) or reducing mode-collapse in GANs (Salimans et al., 2016). We note that we didn't manage to to improve the best attained accuracies by applying layer normalization (Ba et al., 2016) in the LSTM, or adversarial regularization (Miyato et al., 2017) on the word embeddings. Dropout (Srivastava et al., 2014) was by far the most effective form of regularization for the listener ($\sim$[8-9]%), following by $L_2$ weight-regularization on the projected layers ($\sim$[2-3]%).

**Hyper-parameter Search** We did a grid search over the space of hyper-parameters associated with each listener type *separately*. To circumvent the exponential growth of this space, we search it into two phases. First, we optimized the learning rate (in the regime of [0.0001, 0.0005, 0.001, 0.002, 0.004, 0.005]) in conjunction with the drop-out (keep probability) applied at the RNN's input, in the range [0.4-0.7] with increments of 0.05. Given the acquired optimal values, we conducted the second stage of search over the $L_2$ weight-regularization (in the range of [0.005, 0.01, 0.05, 0.1, 0.3, 0.9]), label-smoothing ([0.8, 0.9, 1.0]) and drop-out after vgg/pc-AE projected vectors ([0.4, 0.5, 0.7, 1.0]). In this search, we used a single random seed to control for the data-split which was based on at the object-generalization task.

**Details on the ablated listeners** For the "Separate-Augment", a convolutional layer for aggregating the three encodings showed better performance than an FC. Also, the order-invariant max/mean poolings ($f, g$) produced better results than other alternatives (e.g. using the identity

function in their place). Using a separate MLP to process the point cloud data (via concatenation with the output of the RNN), was slightly better than feeding them directly in the recurrent net (after the tokens of each utterance were processed). However, conditioning the recurrent net with point-clouds and using the images in the end of the pipeline deteriorate *significantly* all attained results. We hypothesize that the gradient flow is better when processing (the inferior in quality) point cloud data, closer to the loss.

**Training details**     We trained the "Proposed" and the "At-Once" for 500 epochs and the ''Separate-Augment" for 350. This was sufficient, as more training only resulted in more overfitting without improving the achieved test/val accuracies. We halved the learning every 50 epochs, if the validation error was not improved during them. Every 5 epochs we evaluated the model on the validation split in order to select the epoch/parameters with the highest attained accuracy. Because the "At-Once" is sensitive in the input order of the geometric codes, we randomly permute them during training. We use the ADAM (Kingma & Ba, 2014) ($\beta_1 = 0.9$) optimizer for all experiments.

## 9.3   SPEAKER DETAILS

| LSTM Size | Learning-rate | $L_2$-reg. | Word-dropout | Image-dropout | RNN-out dropout |
|---|---|---|---|---|---|
| 200 | 0.003 | 0.005 | 0.8 | 0.5 | 0.9 |

Table 5: Optimal hyper parameters for the context-aware neural-speaker.

**Hyper-parameter Search**     To optimize for the hyper-parameters of a speaker we conducted a two-state grid search as for we did for the listeners. First, we optimized (a context-aware speaker with one seed under the object generalization task) with respect to: the hidden neurons of the LSTM (100 and 200), the learning rate ([0.0005, 0.001, 0.003]), the drop-out keep probability for the word-vectors ([0.8, 0.9, 1.0]) and the dropout applied at the RNN's output, before the linear transformation/word-to-logits matrix (with keep probabilities of [0.8, 0.9, 1.0]). The two best models were further optimized by introducing a drop-out layer after the image-projection layer (with keep probabilities in [0.8, 0.9, 1.0]) and $L_2$-weight regularization applied at the same projection layer (with values in [0, 0.005, 0.01]). The optimal parameters of this search are reported in Table 5.

**Model Selection**     To do model selection for a speaker, we used a pre-trained listener (with the same train/test/val splits) which evaluated the synthetic utterances produced by the speaker. To this purpose the speaker generated 1 utterance for each validation triplet via greedy (arg-max) sampling every 10 epochs of training and the listener reported the accuracy of predicting the target given the synthetic utterance. In the end of training (300 epochs), the epoch/model with the highest accuracy was selected.

**Other details**     We initially used GloVe to provide our speaker pretrained word embeddings, as in the listener, but found that it was sufficient to train the word embedding from uniformly random initialized weights (in range [-0.1, 0.1]). We initialized the bias terms of the linear word-encoding layer with the log probability of the frequency of each word in the training data (Karpathy & Fei-Fei, 2015), which provided faster convergence. We train with SGD and ADAM ($\beta_1 = 0.9$) and apply norm-wise gradient clipping with a cut-off threshold of 5.0. The sampled and training utterances have a maximal length of 33 tokens (99th percentile of the dataset) and for each speaker we sample and score 50 utterances per triplet at test time (via Eq. 1). The optimal length penalty ($\alpha$) for the context-unaware speaker is 0.7, and set to 0.6 for the rest.

## 10   PRE-TRAINED IMAGES AND POINT CLOUDS

We train the PC-AE under the Chamfer loss with a bottleneck of 128 dimensions with point clouds of 2048 points extracted from a 3D CAD model, uniformly area-wise. For the VGG-16 encoding, we use the 4096-dimensional output activations of its second fully-connected layer ($fc7$). To fine-tune the VGG-16 we optimized it under the cross-entropy loss for an 8-way classification, which included photo-realistic rendered images of textureless meshes in the 8 largest object classes of Shape-Net (cars, chairs, aeroplanes, ...). The total number of shapes was 36,632. The fine-tuning took 30 epochs of training. The first 15 we optimized only the weights of the last ($fc8$) layer and

the last 15 the weights of all layers. On the test split of a [90%, 5%, 5%] (train/test/val) the network achieves a 96.9 classification accuracy.

## 11 FURTHER QUALITATIVE RESULTS

### 11.1 SPLIT UTTERANCES

While taking entire triplets as input to the listener LSTM did not improve listener performance utterances containing comparatives and superlatives (which in theory should be difficult to evaluate for isolated objects), we also anecdotally considered another subpopulation of utterances that perhaps even more strongly rely on context. These utterances distinguish the target by associating it explicitly with one distractor (e.g. "from the two that have thin legs, the one..."). We used an ad hoc set of search queries to find such utterances among the test set ($\approx 1.5\%$ of utterances) and found that both context-aware architectures do perform noticeably better on these utterances ($67.4 \pm 3.0\%$ for "Separate-Augment" and $65.8\pm5.2\%$ for "At-Once" compared to only $62.5\pm3.7\%$ for the proposed model). However, given the low occurrence of such cases, these effects were not significant and we decided that the negligible gains of the "Separate-Augment" architecture were not worth the increase in model complexity and rigidity with respect to context size (see Section 5.2 for a demonstration of this shortcoming).

Figure 7: **Examples of attention weights on human utterances**. The listener LSTM learns attention weights that emphasize more informative words when forming its linguistic representation. For these speaker utterances drawn from our corpus, we colored each word according to the weight assigned by the attention mechanism, with low attention words in blue and high attention words in red.

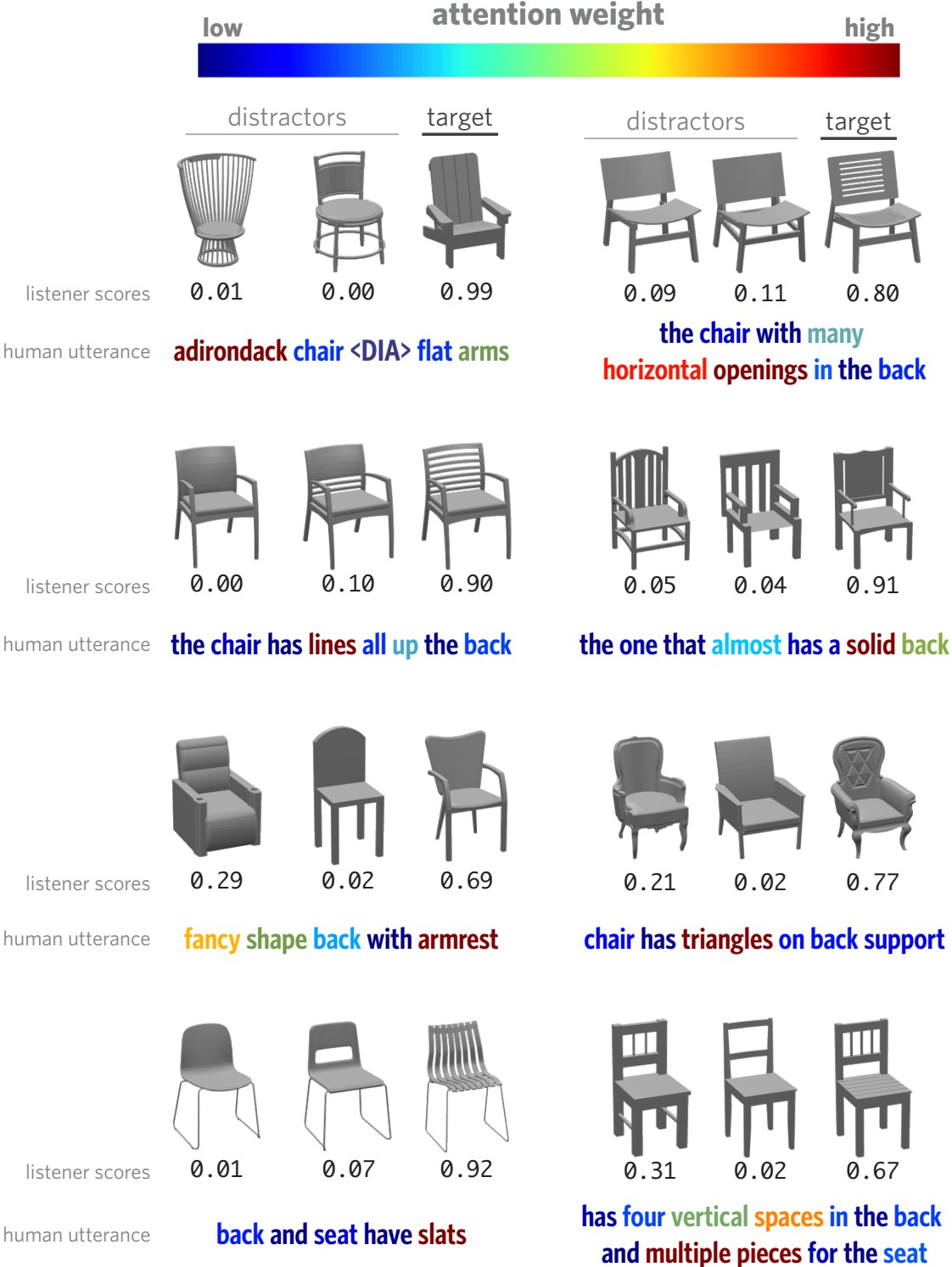

Figure 8: **Examples of errors in listener model**. Our top-performing listener model appeared to struggle to interpret referential language that relied on metaphors, negations, precisely counting parts, ambiguous modifiers, or descriptions of the object's texture or material. All examples are drawn from the test set and were correctly classified by human listeners in the original task.

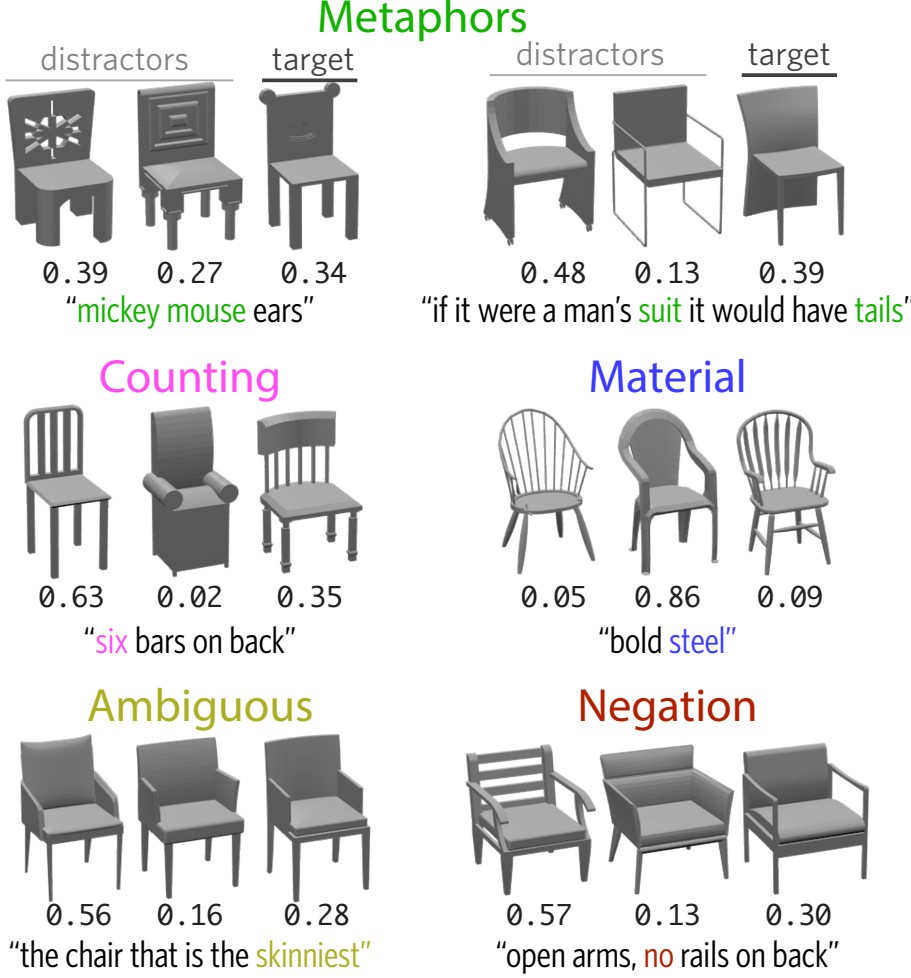

Figure 9: **Examples of errors in speaker models**. Sometimes even the pragmatic (listener-aware) speaker produces insufficiently specific utterances that mention only undiagnostic features, or produces utterances that are literally false of the targert (e.g. there technically *is* a hole in the back) while still succeeding in distinguishing the objects.

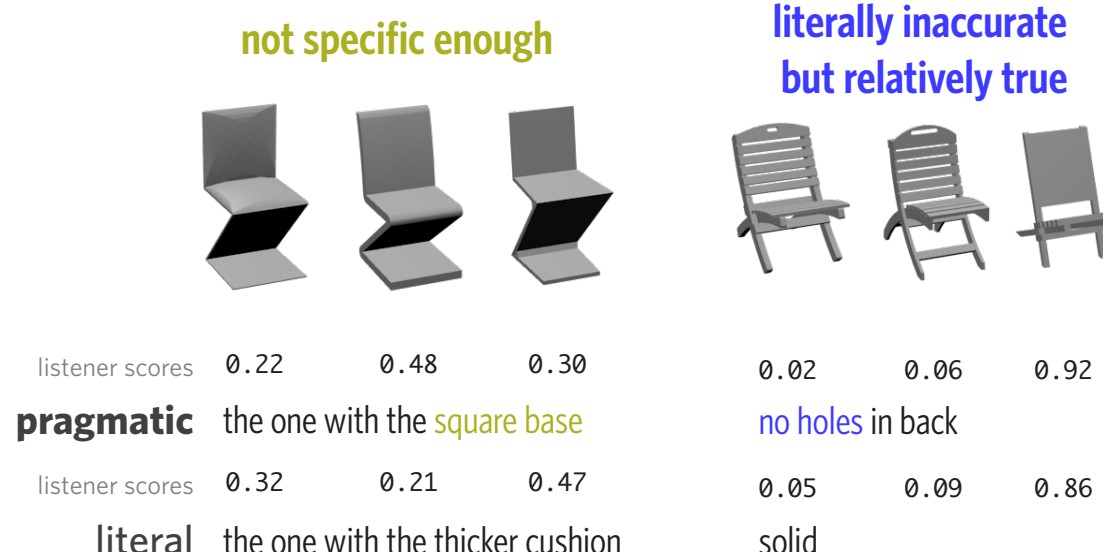

Figure 10: **Search results for chairs** Gallery of retrieved exemplars of held out chairs for different queries. Only the top five are shown.

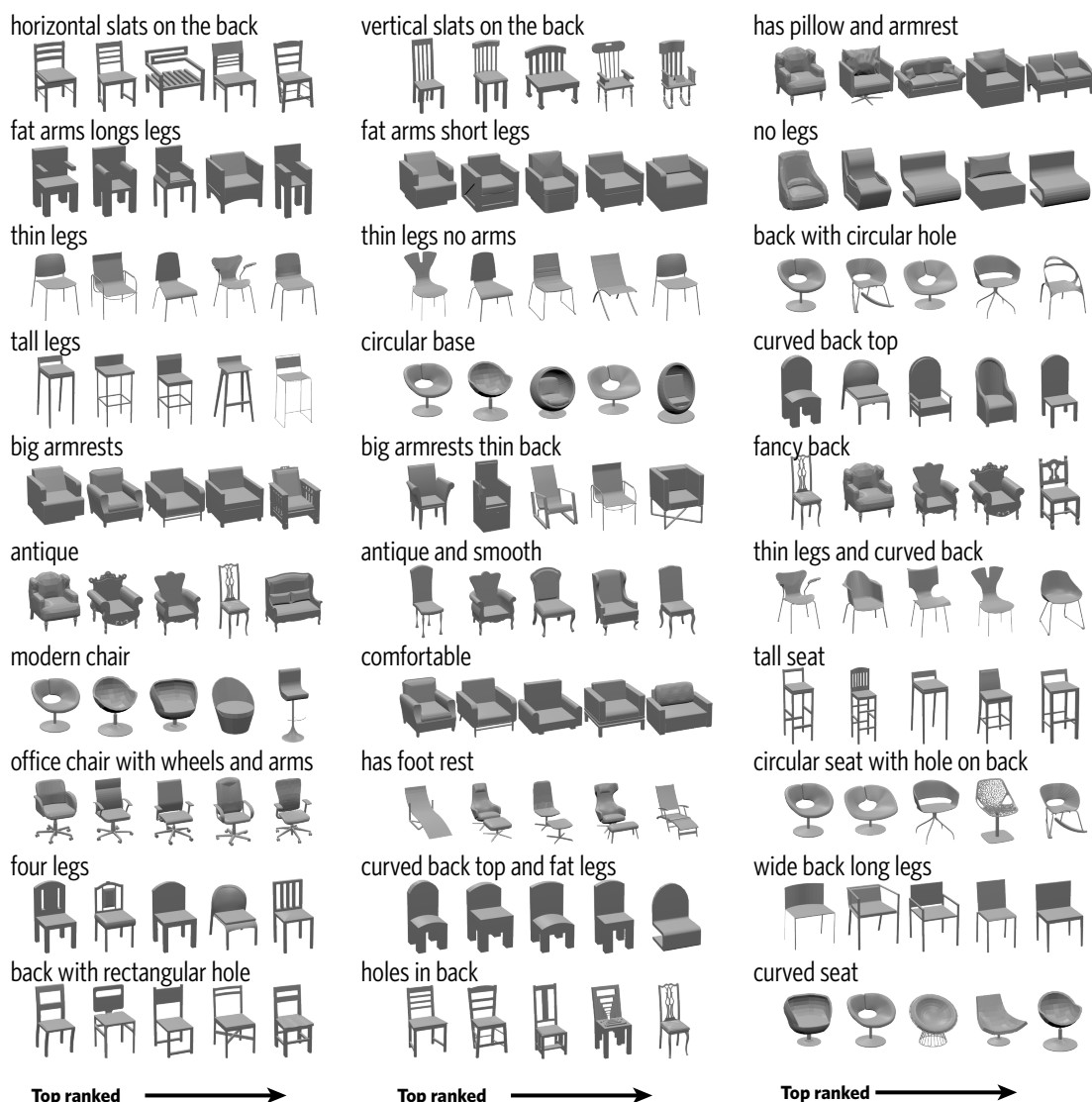

Figure 11: **Search results for out-of-class objects**. Gallery of retrieved exemplars from other ShapeNet furniture categories for different queries. Top five and bottom five are shown, demonstrating intuitive contrasts from the highest ones. Note that there are some mislabeled objects in ShapeNet.

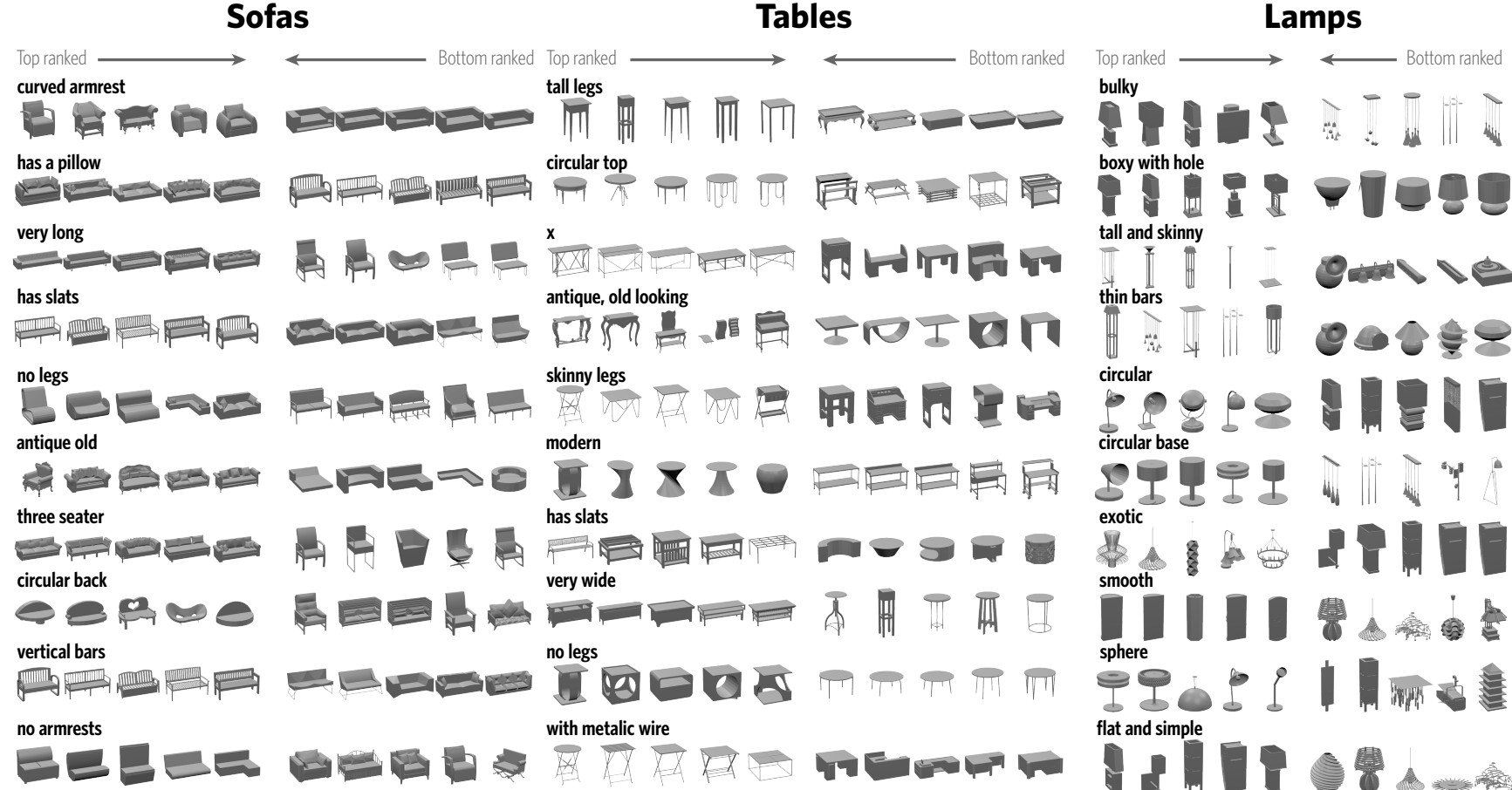

Figure 12: **Effect of context on production**: Synthetic utterances generated by a literal (context-aware) and pragmatic (listener-aware) speaker. The top and bottom rows show utterances produced for the same target in a far and close context, respectively. The best-performing listener's prediction confidence for each object is displayed above: while both speaker models produce similarly effective utterances in far contexts, the literal speaker fails to produce effective utterances in close contexts.

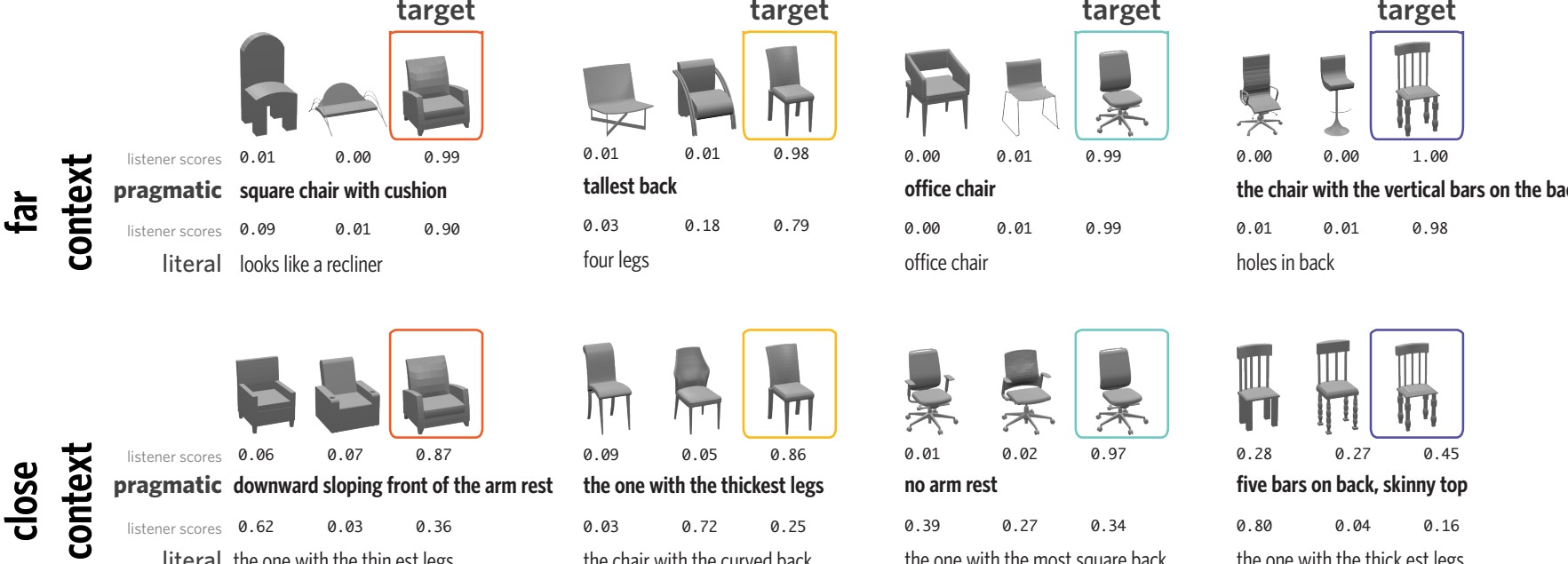

## 12 LENGTH PENALTY AND SPEAKER-AWARENESS

Figure 13: Measuring the effect of using different $\alpha$, $\beta$ values to select the top-1 scoring sentence for context-aware and unaware speakers when creating utterances for the objects/contexts of the validation split. The y-axis in each subplot denotes the performance of a listener who is used to rank *and* evaluate the sentences. Averages are with respect to 5 random seeds controlling the data splits and the initializations of the neural-networks.

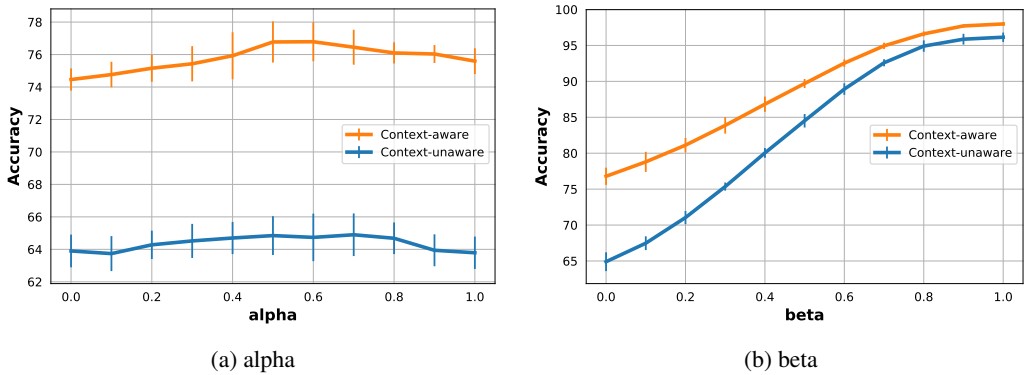

(a) alpha  (b) beta

Figure 14: Effect of using a different fraction of the training data for the evaluating listener when using two separate listeners (for evaluating and scoring a speaker's results.). On the x-axis is the fraction $f$ of the entire training data (80% of the dataset) that is used by the evaluator. $1 - f$ is used by the utterance-scoring listener. Averages are with respect to 5 random seeds controlling the data splits and the initializations of the neural-networks.

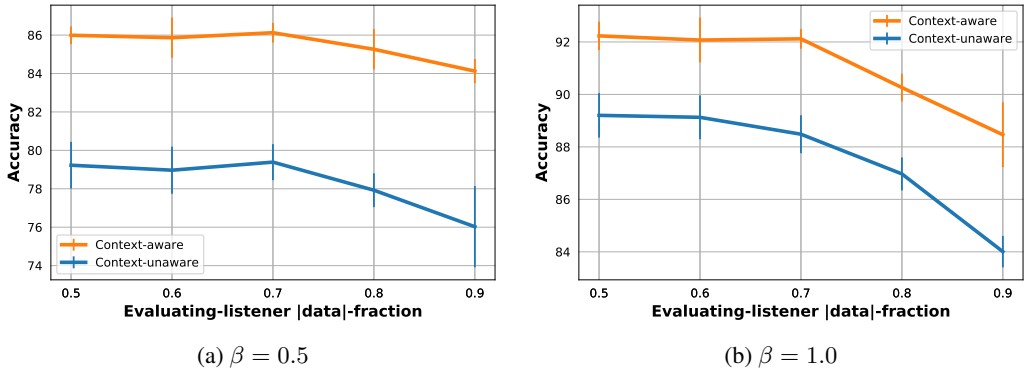

(a) $\beta = 0.5$  (b) $\beta = 1.0$

It is interesting to see that even a context-unaware speaker can generate sentences that a listener can re-rank and then to find the top one as very discriminative. The context-unaware (listener aware) examples in our website demonstrate this improvement.

### 12.1 MORE ABLATIONS

### 12.2 GAME INTERFACE AND CORPUS

Each game consisted of 69 rounds and participants swapped speaker and listener roles after each round. The game's interface is depicted in Figure 16. Participants were allowed to play multiple games, but most participants in our dataset played exactly one game (81% of participants). The most distinctive words in each triplet type (as measured by point-wise mutual information) are shown in Table 7).

Table 6: Performance of different listeners in specific subpopulations in the earlier *language* generalization task. Averages over five random seeds that controlled the data splits and the neural-net initializations.

| Architecture | Subpopulations | | | | |
| --- | --- | --- | --- | --- | --- |
| | **Overall** | **Close** | **Far** | **Sup-Comp** | **Split** |
| Separate (Proposed) | $83.7 \pm 0.2\%$ | $77.0 \pm 0.8\%$ | $\mathbf{90.3} \pm 0.3\%$ | $80.7 \pm 0.3\%$ | $64.6 \pm 3.7\%$ |
| Separate-Augment | $\mathbf{84.4} \pm 0.5\%$ | $\mathbf{78.5} \pm 0.8\%$ | $90.2 \pm 0.7\%$ | $\mathbf{80.9} \pm 0.4\%$ | $\mathbf{68.9} \pm 2.3\%$ |
| Aggregate | $78.4 \pm 0.2\%$ | $71.5 \pm 0.6\%$ | $85.2 \pm 0.3\%$ | $76.0 \pm 0.8\%$ | $61.8 \pm 3.0\%$ |

Figure 15: Listener's accuracy for different sizes of training data, under the *object* generalization task. The original split includes [80%, 10%, 10%] for training/test/val purposes, thus the maximum size of training data is 0.8 of the entire dataset when the fraction is 1.0 (x-axis). The listener model uses the main architecture with using attention, images and point-clouds and its accuracy is always measured on the original (10%) test split. Results with five random seeds controlling the original data split and the neural-net's initialization.

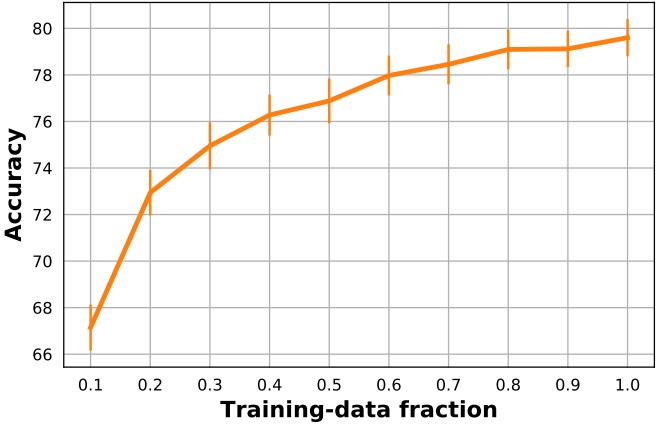

Table 7: Most distinctive words in each triplet type according to point-wise mutual information (excluding tokens that appeared fewer than 30 times in the dataset). Lower numbers are more distinctive of far and higher numbers are more distinctive of close .

| far | **word** | office | sofa | regular | folding | wooden | stool | wheels | metal | normal | rocking |
| --- | --- | --- | --- | --- | --- | --- | --- | --- | --- | --- | --- |
| | **pmi** | -1.70 | -0.94 | -0.88 | -0.84 | -0.83 | -0.79 | -0.78 | -0.71 | -0.67 | -0.66 |
| close | **word** | alike | identical | thickness | texture | darker | skinnier | thicker | perfect | similar | larger |
| | **pmi** | 0.69 | 0.67 | 0.67 | 0.66 | 0.65 | 0.64 | 0.63 | 0.62 | 0.62 | 0.61 |

Figure 16: Reference game interface.

