# OpenReview forum: "Learning to Refer to 3D Objects with Natural Language"
_ICLR.cc/2019/Conference_

### Official Review · AnonReviewer1 · 2018-11-01
**Paper review**

**Rating:** 4
**Confidence:** 4

**Review:**

#update: I've read the authors comments but unfortunately my main concerns about the contributions and novelty of this work are not answered. As such, I cannot increase my score.

------------------

The authors provide a study on learning to refer to 3D objects. The authors collect a dataset of referential expressions and train several models by experimenting with a number of architectural choices.

This is an interesting study reporting results on the effect that several architectural choices have generating referential expressions. Overall, while I appreciate all the experiments and results, I don't really feel I've learned something from this paper.

First and foremost, the paper, from the title already starts to build up expectations about the 3d nature of the study, however this is pretty much ignored at the rest of the paper. I would expect the paper to provide  some results and insights regarding the 3D nature of the dataset and how this affects referential expressions, however, there is no experiment that has used this 3d-ness in any way. Even the representations of the objects are stripped down to essentially 2D (a single-view of a 3D object used to derived VGG features is as 3D as any image dataset used for similar studies, right?).
My major question is then: why should all this research take place in a 3D dataset? Is it to validate that research like this is at all possible with 3D objects?

Moreover, all interesting aspects of referential expressions are stripped out since the authors experiment only with this geometric visual property (which has again nothing to do with 3d-ness, you could totally get that out of images). An interesting study would be to have all objects in the same image and have referential expressions that have to do with spatial expressions, something that the depth or a different view of the of the object could play a role.

Given the fact that there are no technical innovations, I can't vouch for accepting this paper, since there has been quite a lot of research on generating  referential expressions on image datasets (e.g., Kazemzadeh., 2014 and related papers). However, learning to refer to 3D objects is a very interesting topic, and of great importance given the growing interest of training agents in 3D virtual environments, and I would really encourage the authors to embrace the 3d-ness of objects and design studies that highlight the challenges and opportunities that the third dimension brings.


Kazemzadeh et al.: ReferIt Game: Referring to Objects in Photographs of Natural Scenes

---

### Official Review · AnonReviewer2 · 2018-11-03
**an interesting and creative paper**

**Rating:** 6
**Confidence:** 3

**Review:**

The paper investigates how chairs are being described "in the context of other similar or not-so-similar chairs", by humans and neural networks. Humans perceive an object's structure, and use it to describe the differences to other objects "in context". The authors collected a corresponding "chairs in context" corpus, and build models that can describe the "target" chair, that can be used to retrieve the described object, and that can create more discriminative language if given information about the listener.

The paper is well written, in particular the appendix is very informative. The work seems novel in combination with the dataset, and an interesting and well executed study with interesting analysis that is very relevant to both situated natural language understanding and language generation. The "3D" aspect is a bit weak, given that all chairs seem to have essentially been pictured in very similar positions.

---

### Official Review · AnonReviewer3 · 2018-11-04
**Well executed paper, however weak contributions**

**Rating:** 6
**Confidence:** 4

**Review:**

Update: I have read author's response (sorry for being super late). The response better indicates and brings out the contributions made in the paper, and in my opinion is a strong application paper. But as before, and in agreement with R1 I still do not see technical novelty in the paper. For an application driven conference, I think this paper will make a great contribution and will have a large impact. I am slightly unsure as to what the impact will be at ICLR. I leave this judgement call to the AC. I won't fight on the paper in either direction.

The paper studies the problem of how to refer to 3D objects with natural language. It collects a dataset for the same, by setting up a reference game between two people. It then trains speaker and listener models that learn how to describe a shape, and how to identify shapes given a discriminative referring expression. The paper seems to follows state-of-the-art in the design of these models, and investigates different choices for encoding the image / 3D shape, use if attention in the listener, and context and listener aware models.

Strengths:
1. Overall, I think this is a very well executed paper. It collects a dataset for studying the problem of interest, trains state-of-the-art models for the tasks, and conducts interesting ablations and tests insightful hypothesis.

Weaknesses
1. I am not sure what is the technical contribution being made in the paper? Contrastive referential expressions have been used to collect datasets for referring to objects in images. Use of listeners and speakers have been used in NLP (Andreas et al.) as well as in vision and language (Fried et al.). Thus, while I like the application, I am not sure if there is any novel contributions being made in the paper.

Overall, this is a well executed paper, however I am not sure about the novelty of contributions made in the paper.

---

### Meta-Review · Area_Chair1 · 2018-12-17

**Confidence:** 4
**Recommendation:** Reject

**Metareview:**

Paper develops a dataset and model for learning to refer to 3D objects. Reviewers raised concerns about lack of novelty. Fundamentally, it seems unclear what (if any) the take-away for an ML-audience would be after reading this paper. We encourage the authors to incorporate reviewer feedback and submit a stronger manuscript at a future (perhaps a more applied) venue.